# The Use of Commercial Microvolume Techniques for Feline Oocyte Vitrification

**DOI:** 10.3390/ani13010036

**Published:** 2022-12-22

**Authors:** Agnieszka Nowak, Joanna Kochan, Barbara Kij-Mitka, Karolina Fryc, Wojciech Witarski

**Affiliations:** 1Department of Animal Reproduction, Anatomy and Genomics, University of Agriculture, Mickiewicza 24/28, 30-059 Krakow, Poland; 2Department of Animal Nutrition and Biotechnology, and Fisheries, University of Agriculture in Kraków, 24/28 Mickiewicza Ave., 30-059 Krakow, Poland; 3Department of Animal Molecular Biology, National Institute of Animal Production, Krakowska 1 street, 32-083 Balice, Poland

**Keywords:** felids, cats, oocytes, cryopreservation, conservation programs

## Abstract

**Simple Summary:**

This study was conducted to compare the minimum volume vitrification method using material from female felids to protect them in wild felid conservation programs. In the experiment, the three most popular techniques for oocyte vitrification were compared in terms of their survival and developmental competence. The results show that using this method as an emergency protocol is essential to protecting the gametes of wild felids threatened with extinction in conservation programs based on assisted reproductive techniques (ART).

**Abstract:**

This project aimed to compare the three most popular commercial oocyte vitrification techniques to determine their suitability for the vitrification of felid germlines in rescue and conservation programs. The present study aimed to determine the viability and developmental competence of feline oocytes after IVM and vitrification using a commercial vitrification method. In the first experiment, oocytes were vitrified after in vitro maturation (IVM) using the Kitazato, Cryotech, and Vitrolife methods. The oocytes were stained with fluorescein diacetate and ethidium bromide to evaluate their viability. The differences between Vitrolife and the control, Cryotech and Kitazato were statistically significant (*p* < 0.05), and between the control and Kitazato, were highly significant (*p* < 0.01). There were no significant differences between the control and Cryotech, Vitrolife and Cryotech, or Kitazato and Vitrolife. In the second part of the experiment, oocytes, after IVM and vitrification using three commercial methods, were subjected to fertilization. After vitrification, IVF was performed. We observed 35% of embryonic divisions in the group where Vitrolife and Kitazato media were used and 45% in the control group. In the presented experiment, vitrification with Vitrolife media gave slightly better results for survival and fertilization, while in the case of emergency protocol vitrification, all of the above methods may be useful to protect material derived from valuable wild felids.

## 1. Introduction

Vitrification is an essential tool in assisted reproduction programs, allowing preservation of the genetic material of endangered species and valuable individuals for breeding and companion animals [1,2,3,4,5,6,7]. Vitrification improves cryopreservation outcomes and, therefore, the success rate of IVF using frozen embryos and gametes [8]. From an experimental point of view, it has become a method that allows researchers, in many cases, to obtain results comparable to those of fresh oocytes [8].

The latest advances in cryobiology have led to newer options or protocols for oocyte cryopreservation in terms of techniques, the composition of cryoprotectants, and various devices, resulting in improved survival of cryopreserved oocytes. Ultra-fast vitrification methods allow oocytes to “safely” pass through the critical temperature zone from 0 °C to −15 °C so quickly that no ice crystals can form that could damage the cell. It has become common to replace plastic straws with other devices, such as closed, pulled OPS straws [9], a cryoloop [10], nylon mesh [11], or a Cryotop [12,13,14]. All of them make it possible to obtain cooling rates above −100,000 °C/min.

Storing oocytes at liquid nitrogen temperature is of practical, economical, and ethical importance. It is also used in programs that support animal reproduction. The cryopreservation of oocytes obtained from live females saves valuable genetic material and eliminates the need for immediate the fertilization of oocytes. Using the oocyte vitrification technique in livestock breeding programs will enable better management of genetic material, improve the population, and indirectly regulate the number of births in a given population. In the case of threatened and endangered species, it can lead to an increase in the population by using oocytes in assisted reproductive techniques and strengthening the population’s genetic material. Thanks to cryopreservation, it is possible to reduce the costs and risks associated with transporting breeding animals [15]. As with other mammals, poor genetic diversity is one of the biggest problems in non-domestic cat populations. Therefore, it is essential to secure genetic material from a representative number of animal populations and store it to preserve its vitality for the future [7,8,16,17,18,19,20]. Therefore, it is crucial to develop and adopt a method that will make it possible to protect material derived from genetically valuable individuals. In many cases, this protection is an emergency, so this method must be easy to perform while ensuring a high rate of oocyte survival and comparable fertilization and embryo results, as in the case of fresh oocytes.

The aim of this work is to compare the minimum volume methods for the vitrification of felid oocytes.

## 2. Materials and Methods

The media used for IVF, GM501 Wash medium and GM501 Cult medium, are commercial media for human IVF (Gynemed, Lensahn, Germany). Unless stated otherwise, the remaining chemicals and reagents used in this study were purchased from Sigma-Aldrich (St. Louis, MO, USA).

### 2.1. Study Design

#### 2.1.1. Experiment I: Evaluation of the Viability Rate for Oocytes

In this experiment, three different media were evaluated. The first batch of oocytes were pooled in the following groups:-Control IVM (CONT/IVM);-IVM + vitrification with Cryotech media (CRYOTECH);-IVM + vitrification with Kitazato media (KITAZATO);-IVM + vitrification with Vitrolife media (VITROLIFE).

#### 2.1.2. Experiment II: Evaluation of the Impact of the Oocyte Vitrification Procedure for the Development of Embryos after IVF

The second batch of oocytes were pooled in the following groups:-Control IVF (CONT/IVF);-Vitrification with Cryotech media + IVF (CRYOTECH/IVF);-Vitrification with Kitazato media + IVF (KITAZATO/IVF);-Vitrification with Vitrolife media +IVF (VITROLIFE/IVF).

### 2.2. Ovary Collection

Oocytes for in vitro culture were obtained from ovaries collected from healthy, sexually mature females during routine ovariohysterectomy, conducted at local veterinary clinics in Cracow. Ovaries were kept in DPBS with 100 ug/mL streptomycin and 100 IU/mL penicillin at 4 °C. Oocytes were obtained within 1–3 h after ovariohysterectomy.

### 2.3. Oocyte Collection and In Vitro Maturation of Oocytes

Cumulus–oocyte complexes (COCs) were obtained through the scarification method in a washing medium (GM501 Gynemed, Lensahn Germany).

Oocytes with dark cytoplasm and compact layers of cumulus cells were selected for IVM, and placed into 4-well dishes with approximately 30 COCs per well in 400 mL BO-IVM medium (IVF Bioscience, Sokolow Podlaski, Poland) [21]. Incubation took place for 24 h at 37 °C and 5% CO_2_. After IVM, oocytes were denuded enzymatically (hyaluronidase solution (GM501 Hyaluronidase, Gynemed, Germany)) for five min, and mechanically by pipetting them into a washing medium (GM501 Wash medium with phenol red and Gentamicin, Gynemed, Germany). Oocytes with a visible first polar body (metaphase II) were used for further procedures.

### 2.4. Vitrification and Warming of Oocytes

Three vitrification and warming protocols were used: Cryotech, Kitazato, and Vitrolife (Table 1).

#### 2.4.1. Cryotech Method Protocol

Vitrification was performed using Cryotech vitrification media (Cryotech, Japan) in two temperature steps of 25 and 27 °C. The oocytes were equilibrated for 15 min in equilibration solutions (ES), and then, put into vitrification solution (VS) for 1.5 min, and immediately placed into the device and plunged in liquid nitrogen. During warming, the Cryotech device was removed from the liquid nitrogen and put into pre-warmed (37 °C) thawing medium (TS) for 1 min; then, it was put into a diluent solution (DS) for 3 min, and next, into washing solution (WS) for 5 min.

#### 2.4.2. Kitazato Method Protocol

Vitrification with Kitazato was performed (at temperatures of 25 and 27 °C). The oocytes were equilibrated for 15 min in ES, and then, transferred to the VS for 1.5 min and placed in a cryo-device immersed in liquid nitrogen. To warm the oocytes, the device was removed from the liquid nitrogen and immersed into the warmed (37 °C) TS medium for 1 min; after that, it was put into a DS medium for 3 min, and into WS for 5 min.

#### 2.4.3. Vitrolife Method Protocol

Vitrification was conducted using Vitrolife vitrification media and Rapid-I devices (Vitrolife) at a temperature of 37 °C according to the manufacturer’s protocol. The oocytes were put in Vitri 1 for 5–20 min; then, they were transferred to Vitri 2 medium for 2–5 min, and then, to the last droplet: Vitri 3. The total exposure time, from the oocytes entering Vitri 3 until they were put on the Rapid-I device and underwent vitrification, lasted 25–35 s. The warming process started with the removal of the Rapid-I device from the Rapid straw (remaining in liquid nitrogen); then, it was immersed into Warm 1 medium for 1 min, and then, put into a Warm 2 medium for 3 min, Warm 3 for 5 min, and finally, Warm 4 medium for 5-10 min.

Oocytes, after the vitrification/warming procedures, were transferred into PBS solution, and then, evaluated via a viability test to IVF.

### 2.5. Viability Test of Oocytes

Oocytes were put on a glass slide in a DPBS; (Sigma-Aldrich, St. Louis, MO, United States) drop containing 0.05 mg ml^−1^ ethidium bromide (EtBr excitation peak: 518 nm, emission: peak 605 nm) and 0.005 mg ml^−1^ fluorescein diacetate (FDA excitation peak: 498 nm, emission peak: 517 nm). The evaluation was performed using a Nikon Eclipse E600 fluorescence microscope and 200× magnification. Oocytes with green fluorescence were assessed as live, while those which demonstrated red-orange fluorescence were classified as dead.

### 2.6. In Vitro Fertilization

Thawed spermatozoa isolated from the cauda epididymis and frozen according to the procedure described by Niżański et al. [22] were used. After thawing (37 °C for 30 s.), the swim-up method was used (Sperm Air^®^ Gynemed, Lensahn, Germany). Oocytes were inseminated (5 × 10^5^ motile spermatozoa/mL in 50µL micro drops of Cult^®^ Medium (Gynemed, Lensahn, Germany). The IVF procedure was performed at 38.5 °C in air with 5% CO_2_. Oocytes and spermatozoa were cultured for 16 h.

### 2.7. In Vitro Zygote/Embryo Culture

After IVF, the presumptive zygotes were cultured in GM501 Cult medium with Gentamicin and Phenol red (Gynemed) with mineral oil (GM501 Mineral oil Gynemed) at 38.5 °C in a 5% CO_2_ atmosphere, and evaluated every 24 h according to Gardner DK, Schoolcraft [23].

### 2.8. Statistical Analysis

The statistical analysis was conducted using the statistical package STATISTICA version 9.1 (StatSoft, Inc., Tulsa, OK, United States 2010). The difference between the groups was considered significant when *p* < 0.05 and highly significant when *p* < 0.01. Non-parametric data were compared using a chi-square test.

## 3. Results

A total of 68 pooled ovaries from 34 animals were utilized in this study, from which 744 oocytes were collected. Only 560 oocytes with dark cytoplasm were used for IVM. After IVM, 280 oocytes (47%) had a visible first polar body. In the first part (Experiment I), 200 oocytes in metaphase II (MII) were used, whereas in the second part (Experiment II), 80 oocytes were used.

### 3.1. Experiment I

Table 2 presents data on the viability test after vitrification (Experiment I). After the vitrification process, 200 matured oocytes were used for viability evaluation. No statistically significant differences between the analyzed groups were observed, but of all the experimental groups evaluated, the best percentage of survival rate was obtained in the group where oocytes, after in vitro maturation, were vitrificated by Vitrolife media: the IVM + vitrification + Vitrolife media (VITROLIFE) group. In this group, the percentage of oocytes classified as “alive” was 84.5%. Similar results were obtained in the remaining experimental groups: IVM + vitrification + Cryotech media (CRYOTECH) and IVM + vitrification + Kitazato media (KITAZATO), respectively: 84% and 79.6%.

### 3.2. Experiment II

The IVF procedure was performed after the vitrification/thawing of 80 oocytes. Twenty oocytes from each vitrified/warmed group were submitted to IVF. The control group consisted of twenty oocytes submitted to IVM and not vitrified. After fertilization, the presumptive zygotes were cultured. The results are presented in Table 3. We observed that 35% of embryonic divisions were found in the IVF group where Vitrolife (vitrification + Vitrolife media + IVF (VITROLIFE/IVF)) and Kitazato media (vitrification + Kitazato media + IVF (KITAZATO/IVF)) were used, and 45% were found in the control group (Control IVF (CONT/IVF)).

## 4. Discussion

The cryopreservation of oocytes is a method of retaining useful genetic material. Gamete rescue from wild females is often required after medically indicated castration, euthanasia, or sudden death [24]. Vitrification is cheap and easy and produces less ultrastructural damage than slow freezing [25]. In the previous study, we confirmed that domestic and wild cat oocytes could survive vitrification using an ultrarapid, minimum-volume cooling technique [7,14]. However, it is essential for other individuals to use the most effective vitrification method. Therefore, the authors attempted to compare the three most popular methods of vitrification in terms of effectiveness and ease of implementation of the procedure to optimize the protocol for felids. From a technical point of view, in the case of a person with good laboratory skills, working with these three devices is easier; therefore, differences in the ease/difficulty of working with these devices in the presented publication were not analyzed.

Gamete banking may be the optimal tool when working with species where only a few individuals remain per population and genetic diversity is hugely threatened [6]. In their paper, Zahmel et al. [26] vitrificated lion oocytes after collection, and then, used IVM protocols. The maturation rates were 55% and 49.2% for the control and vitrified group, respectively. In another paper, the maturation percentage after the vitrification of immature cumulus–oocyte complexes (COCs) was approximately 22–37%, while the cleavage percentages dropped to 6–24% [16,18,27,28,29]. In our papers, we decided to perform vitrification the oocytes after maturation. The percentage of matured oocytes before vitrification without cumulus cells was 47%. In the case of micro-volume methods, the rapid penetration of cryoprotectants inside the oocyte is required. Immature oocytes are surrounded by cumulus cells, which may be a physical barrier for cryoprotectants [16]. Cocchia et al. analyzed the viability of cat oocytes vitrified in OPS at the GV stage using cFDA/Tripan blue staining [18]. In this paper, viable COCs amounted to 45.3% after vitrification. [18]. These results are similar to ours, where oocytes were vitrified after maturation. From a technical point of view, it is difficult to compare both methods because the OPS method is not included in the microvolume methods. However, in the case of wild felines, all protocols should be considered appropriate to best preserve the material, technical skills, and equipment availability.

In humans and livestock animals, commercial sets of vitrification are increasingly used to facilitate work and obtain repeatability of the compositions and concentration [6,7,30]. In our experiment, the best vitrification efficiency (the highest percentage of viable oocytes) was noted when using the Cryotech and Vitrolife methods. The percentages of viable oocytes were 84% and 84.5%, respectively (Table 2). Our previous study [7] used the Vitrolife method for the vitrification of wild felids oocytes. We reported 70% of viable serval oocytes and 60% of viable Pallas’s cat oocytes after vitrification [7]. Fernandez-Gonzalez and coauthors compared three protocols for the vitrification of immature-stage oocytes (two commercial kits, with a three-step method for the vitrification of cat oocytes [25]). The Cryotop method showed the lowest maturation percentage obtained after warming (10.1%). A significant difference in the maturation percentage of oocytes was found between Kitazato (38.7%) and the three-step method (24.5%) [25]. In the case of the Vitrolife method, it should be noted that the procedure is performed at 37 °C, while the other two methods are at 25 °C (Table 1). This helps to maintain the spindle integrity and viability of oocytes and embryos. From a practical point of view, there is no need to lower the working surface temperature during vitrification. A paper by Mikołajewska et al., shows that the vitrification solutions consisted of 20% ethylene glycol, 20% DMSO, 20% FCS, and 1.5 m Trehalose with and without 10% Ficoll PM-70. The authors demonstrated 52% and 41% of live mature and immature feline oocytes after vitrification. Working at physiological temperatures shortens the vitrification time and minimizes the exposure of embryos to cryoprotectants and their potential toxic effects [31].

Survival rates from mature vitrified and warmed cat oocytes of 52% have been described [8]; in contrast, in human and mouse species, 96% was reported. [32,33]. In 1997, Luvoni [34] reported the cryopreservation of feline oocytes for the first time and her study indicated the superiority of the slow-freezing method. Vitrified oocytes are subjected to IVF or assisted fertilization after thawing. Therefore, the vitrification procedure must not disturb the ability to fertilize oocytes. In a previous report, we used parthenogenetic activation to evaluate the development competence of vitrificated cat oocytes [14]. In vitrification by the Cryotop group, 46% cleaved embryos were obtained [14]. Selecting the stage at which oocytes will be vitrified affects their survival, fertilization, and development rate. In their paper, L Fernandez-Gonzalez and K Jewgenow [25] found that the cleavage after ICSI of warmed and matured oocytes was 20.7% (Kitazato method) and 28.6% (hand-made protocol), and the morula percentage was 18.2% and 22.5%, respectively; however, they did not reveal any significant differences between the two methods. In our study, oocytes vitrified after IVM were fertilized with classic IVF. In the three experimental groups, the cleavage rate was obtained at 35% (Cryotech and Vitrolife media) and 25% (Kitazato media). No statistical differences were observed in the obtained blastocysts depending on the vitrification method used. In their paper, Sowińska et al. [35] found that oocyte status before vitrification (immature vs. in vitro matured) did not influence fertilization and morula rates (*p* > 0.05). However, the cleavage rates were significantly reduced between the two vitrification groups (2.6%, 13.5%) and the non-vitrified group (28.3%). In the mentioned publication, the authors used the ICSI technique, which, compared to IVF, is much more effective.

## 5. Conclusions

Domestic cats are model animals for wild species. Vitrification plays a crucial role in endangered species conservation programs. However, based on available publications and the presented results in this paper, it is difficult to choose the most effective protocol.

Using commercial protocols for vitrification, the in vitro maturation of oocytes, and fertilization and embryo culture offers many opportunities for a cheap, fast, and reproducible method to preserve the material. However, it still leaves many questions, doubts, and inconclusive results. The protocol choice should be adapted to a specific situation, and further research is necessary to protect this unique material.

## Figures and Tables

**Table 1 animals-13-00036-t001:** The comparison of media composition and vitrification/thawing protocol between experimental groups (according to the manufacturers’ leaflets).

	Cryotech	Kitazato	Vitrolife
Temperature of vitrification	25 °C	25 °C	37 °C
Temperature of warming	37 °C (1 min), then, 25 °C	37 °C (1 min), then, 25 °C	37 °C
Media composition	Modified HEPES Buffered (MEM) Hydroxypropyl cellulose (HPC) Ethylene glycol (EG) Dimethyl sulfoxide (DMSO) Endotoxin-free trehalose	HEPES buffer Dimethyl sulfoxide (DMSO) Ethylene glycol (EG) Trehalose Hydroxypropyl cellulose (HPC) Gentamicin	MOPS-buffered medium Human serum albumin Ethylene glycol (EG) Propanodiol Sucrose Gentamicin
Device	Flat device (Cryotech device)	Flat device (Kitazato device)	Rapid-I device with hole

**Table 2 animals-13-00036-t002:** Results of vitrification of IVM oocytes with different commercial protocols.

Liveliness Evaluation	CONT/IVM ^a^	CRYOTECH ^ab^	KITAZATO ^c^	VITROLIFE ^bc^
Vitrification oocytes	-	50	50	50
Thawing oocytes	-	50	49	45
		(100%)	(98%)	(90%)
Live oocytes *n* (%)	50	42/50	39/49	38/45
	(100%)	(84%)	(79.6%)	(84.4%)
Dead oocytes *n* (%)	0	8/50	10/49	7/45
		(16%)	(20.4%)	(15.6%)

The same letters indicate no statistically significant differences between analyzed groups, unlike (a vs. b) and (a vs. c), both with *p* < 0.001, and (b vs. c), with *p* < 0.05. IVM—in vitro maturation; CRYOTECH—Cryotech media protocol for vitrification/thawing; KITAZATO—Kitazato media protocol for vitrification/thawing; VITROLIFE—VitOmni media protocol for vitrification/thawing.

**Table 3 animals-13-00036-t003:** Results of IVF.

Medium	Number of Oocytes	CR-Cleavage Rate *	2–8 Blastomere-Stage Embryos **	Morula Stage	Blastocyst Stage	No Division ***
CONT/IVF ^a^	20	9/20	4/9	1/9	4/9	11/20
	(100%)	(45%)	data	(11%)	(45%)	(55%)
CRYOTECH/IVF ^a^	20	5/20	2/5	2/5	1/5	15/20
	(100%)	(25%)	(40%)	(40%)	(20%)	(75%)
KITAZATO/IVF ^a^	20	7/20	3/7	3/7	1/7	13/20
	(100%)	(35%)	(43%)	(43%)	(14%)	(65%)
VITROLIFE/IVF ^a^	20	7/20	3/7	2/7	2/7	13/20
	(100%)	(35%)	(29%)	(29%)	(29%)	(65%)

The same letters indicate no statistically significant differences between analyzed groups. * CR—Cleavage rate (percentage of cleaved embryos assessed at Day 1 relative to the number of matured oocytes). ** Embryos that stopped developing at cleavage stage (2- to 8-blastomere stage). *** No division—percentage of oocytes without division (no fertilization of oocytes).

## Data Availability

The data presented in this study are available in this paper. Detailed data on the wild felids used in this study (CITES reports) are available on request from the corresponding author.

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
