# Peer review of "The Use of Commercial Microvolume Techniques for Feline Oocyte Vitrification"

_animals, 2022, doi:10.3390/ani13010036_

Round 1

Reviewer 1 Report

General comments: the manuscript entitled “The use of commercial microvolume techniques for feline oocyte vitrification” aimed to compare 3 commercial vitrification kits to cryopreserve feline oocytes and to evaluate embryo development. The study is simple and it used a low number of oocytes, specially when evaluating developmental competence. English is not bad, however there are some minor grammar mistakes as well as some mistyping.  I believe there are several points to be improved before the manuscript is considered for publication. 

Specific comments:

Abstract: it should include p values. Keywords should not repeat words that are already in the title.

L31: “after vitrification, ICSI was performed…” but on M&M authors described IVF procedure. Which one was used?

L32-33: please correct the sentence.

L73-74: please review this sentence.

M&M: As the authors are doing vitrification, use “Warming” instead of “thawing”.

L81: please review this sentence.

L83-86:” IVM+vitrification with cryotech media” instead of “IVM+vitrification+cryotech media”. Same for the other groups.

L90-92: same comment as the above.

L105: “CO2” please correct the subscript.

L112: Tables should be mentioned in the order of appearance in the text and so this should be table 1.

L145: please include wavelength and emission. Please include magnification.

L153: “5 x 105” please correct the superscript.

L154 and 159: “CO2” please correct the subscript.

L163-164: Please change to “The difference between the groups was considered significant for p<0.05 and highly significant for p<0.01.”.

L172-173: there is no need to repeat the information on vitrification methods.

L174-179: I believe these sentences should be revised once there is no significant difference between cryotech and vitrolife on survival rates. From a statistical point of view, we cannot assume that “the best results were obtained in vitrolife…”. 

Percentage should be written “84.4%” and not with a comma (84,4%) and should be equal in the text and in the table (79.6% in the text vs. 80% in the table).

L188: “warming” instead of “thawing”. Please replace the sentence “Sixty oocytes were vitrified…” by “Twenty oocytes from each vitrified/warmed group were submitted to IVF”.

L189: Please review the sentence. I suggest something like “Control group consisted of twenty oocytes submitted to IVM and not vitrified”.  

L192: “Where” instead of “were”.

Discussion: 

L203-214: This paragraph is more appropriate for introduction rather than discussion. 

L216-230: This paragraph just presents general ideas of the choice to vitrified mature oocytes. It should be revised focusing on the results of the study.

L231-236: Again, this paragraph looks more like an introduction or literature review.

L240-244: it is a repetition of the results.

L244-254: There was no significant difference between oocyte viability in the Cryotech and Vitrolife groups and so, this explanation is not quite precise. 

L256-262: No result is being discussed here.

L263-272: Authors should link this information with their own data otherwise it looks like a review.

L273-279: This is a repetition of the results.

Conclusions: It should include an input of the results obtained in the study, not an overall statement. 

Table 1: Authors said 200 matured oocytes were used for viability evaluation, but there are 190 oocytes on the table (50 for each vitrification method and 40 for control group).

Table 2: in the column “2-8 blastomeres embryos”?data? 

First column: No oocytes stand for “number of oocytes” while in the last column “No division” stands for “no division”, correct? I suggest correcting it to clarify this misunderstanding. 

There are no letters indicating significant differences. 

Table 3: should be the first table.  Title and 1st column: “warming” instead of “thawing”. 

Author Response

Dear Sir/Madam

Thank you for the review.

Corrections were applied as suggested, and we hope you will find the modified manuscript entirely acceptable for publication. We are very grateful for the comments, which have helped us improve the quality and impact of the manuscript.

The answers are provided below:

Specific comments:

Abstract: it should include p values. Keywords should not repeat words that are already in the title.

Thank you for the suggestions. Comments have been included in the manuscript.

L31: "after vitrification, ICSI was performed…" but on M&M authors described IVF procedure. Which one was used?

We use IVF in this paper. The stylistic mistake was corrected in the text. Thank you for the suggestions.

L32-33: please correct the sentence.

L73-74: please review this sentence.

L81: please review this sentence.-

Thank you for the suggestions. The sentence was changed.

M&M: As the authors are doing vitrification, use "Warming" instead of "thawing".

Thank you for the suggestions. Comments have been included in the manuscript.

L83-86:" IVM+vitrification with cryotech media" instead of "IVM+vitrification+cryotech media". Same for the other groups. And L90-92: same comment as the above.

Thank you for the suggestions. Comments have been included in the manuscript.

L105: "CO2" please correct the subscript.

Thank you for the suggestions. Comments have been included in the manuscript.

L112: Tables should be mentioned in the order of appearance in the text and so this should be table 1.

Thank you for the suggestions. Comments have been included in the manuscript.

L145: please include wavelength and emission. Please include magnification.

Thank you for the suggestions. Added to the M&M part information.

L153: "5 x 105" please correct the superscript.

Thank you for the suggestions. Comments have been included in the manuscript.

L154 and 159: "CO2" please correct the subscript.

Thank you for the suggestions. Comments have been included in the manuscript.

L163-164: Please change to "The difference between the groups was considered significant for p<0.05 and highly significant for p<0.01.".

Thank you for the suggestions. Comments have been included in the manuscript.

L172-173: there is no need to repeat the information on vitrification methods.

Thank you for the suggestions. Comments have been included in the manuscript.

L174-179: I believe these sentences should be revised once there is no significant difference between cryotech and vitrolife on survival rates. From a statistical point of view, we cannot assume that "the best results were obtained in vitrolife…".

Thank you for the suggestions. We added to the manuscript information about no statistical difference and corrected the sentence about the percentage of survival rate.

Percentage should be written "84.4%" and not with a comma (84,4%) and should be equal in the text and in the Table (79.6% in the text vs. 80% in the Table).

Thank you for the suggestions. Comments have been included in the manuscript.

L188: "warming" instead of "thawing". Please replace the sentence "Sixty oocytes were vitrified…" by "Twenty oocytes from each vitrified/warmed group were submitted to IVF".

Thank you for the suggestions. Comments have been included in the manuscript.

L189: Please review the sentence. I suggest something like "Control group consisted of twenty oocytes submitted to IVM and not vitrified". 

Thank you for the suggestions. Comments have been included in the manuscript.

L192: "Where" instead of "were".

Thank you for the suggestions. Comments have been included in the manuscript.

Discussion:

L203-214: This paragraph is more appropriate for introduction rather than discussion.

Thank you for the suggestions. The paragraph has been retained, but as suggested to be appropriate for an introduction, it has been shortened to be a minor introduction to the discussion.

L216-230: This paragraph just presents general ideas of the choice to vitrified mature oocytes. It should be revised focusing on the results of the study.

Thank you for the suggestions. General ideas from this paragraph were delayed. As suggested, a comparison with the results of other authors has been added.

L231-236: Again, this paragraph looks more like an introduction or literature review.

Thank you for the suggestions. The paragraph was removed from the Discussion part.

L240-244: it is a repetition of the results.

The paragraph was corrected. The results were left and compared with other publications. Thank you for the suggestions.

L244-254: There was no significant difference between oocyte viability in the Cryotech and Vitrolife groups and so, this explanation is not quite precise.

The paragraph has been redrafted to correctly present the advantage of one of the commercial sets as easier to use. In the case of emergency protection of the material, from the authors' perspective, it is easier, from a practical point of view, to operate in the set of one temperature only. Therefore, we wanted to add this information to this paragraph.

L256-262: No result is being discussed here.

The paragraph has been shortened to contain only information relevant to the publication. Thank you for the suggestions.

L263-272: Authors should link this information with their own data otherwise it looks like a review.

Thank you for the suggestions. Comments have been included in the manuscript.

L273-279: This is a repetition of the results.

Thank you for the suggestions. The results presented in the publication were compared with the works of other authors.

Conclusions: It should include an input of the results obtained in the study, not an overall statement.

Thank you for the suggestions. Comments have been included in the manuscript.

Table 1: Authors said 200 matured oocytes were used for viability evaluation, but there are 190 oocytes on the Table (50 for each vitrification method and 40 for control group).

The mistake was corrected. Thank you for the suggestions.

Table 2: in the column "2-8 blastomeres embryos"?data?

The column included embryos in stage from 2 to 8 blastomeres. All embryos in this stage were pooled in this column. The information was added to the table 3.

First column: No oocytes stand for "number of oocytes" while in the last column "No division" stands for "no division", correct? I suggest correcting it to clarify this misunderstanding.

Thank you for the suggestions. The column titles were corrected to clarify the matter.

There are no letters indicating significant differences.

Thank you for the suggestions. The letters indicating significant differences are added to the name of the experimental group.

Table 3: should be the first Table. Title and 1st column: "warming" instead of "thawing".

Thank you for the suggestions. Comments have been included in the manuscript.

Reviewer 2 Report

This article compares different vitrification methods for conservation of female fertility especially in endangered species. This is an important endeavour since knowing which vitrification procedure works for different species is important for future IVF purposes. 

The authors compare 3 methods out of which one is a bit better but most are comparable in the statistics. I think this will be useful information for researchers and clinicians alike who wish to use these kits. But some of the information for the experiments are confusing and require clarification. The English and grammar also require careful proofing and checks. Otherwise, the claims are supported by the observations. 

Here are some points to consider:

1. The main results sentence in the abstract (line 32-33) did not make much sense and needs to be rewritten with proper grammar to be understood. This is important since readers should not be guessing outcomes of the experiments in the abstract. It should be mentioned clearly. 

2. In the materials and methods section it should be made clear which cohort of oocytes were used for freeze/thaw/IVF cycles. It was unclear whether the oocytes from the first table were the same ones in the second. If not, then how long were the oocytes vitrified and frozen for? This would appear to be a really important consideration given this method will be used after storing oocytes for a while. This should be made clear in the methods.

3. The oocytes are matured to meiosis II eggs I presume?

4. Can the authors comment on the ease of the device used in the vitrification methods? Is the Rapid-I device different from the flat ones? Is it easier to use?

5. Please provide info about what the numbers indicate in the tables. If oocyte numbers then please indicate from how many animals/hysterectomy procedures these are collected from. Otherwise, the statistics is hard to evaluate.

Minor points: 

1. Please check English grammar. This is a minor point but will greatly help understanding the material. 

2. Please proofread to make sure periods are after every sentence, for e.g. Line 52.

3. Line 82- I think the authors mean “pooled” and not “pulled”.

4. Line 117 – Should be “plunged” and not “plugged”. 

Author Response

Dear Sir/Madam

Thank you for the review.

Corrections were applied as suggested, and we hope you will find the modified manuscript entirely acceptable for publication. We are very grateful for the comments, which have helped us improve the quality and impact of the manuscript.

The answers are provided below:

Specific comments:

This article compares different vitrification methods for conservation of female fertility The main results sentence in the abstract (line 32-33) did not make much sense and needs to be rewritten with proper grammar to be understood. This is important since readers should not be guessing outcomes of the experiments in the abstract. It should be mentioned clearly. 

Thank you for the suggestions. Comments have been included, and the sentence was rewritten.

  1. In the materials and methods section it should be made clear which cohort of oocytes were used for freeze/thaw/IVF cycles. It was unclear whether the oocytes from the first Table were the same ones in the second. If not, then how long were the oocytes vitrified and frozen for? This would appear to be a really important consideration given this method will be used after storing oocytes for a while. This should be made clear in the methods.

Thank you for the suggestions. Comments have been included and the sentence was rewritten. We added in Study design information "the first part of oocytes and  The second part of oocytes" to note that these are separate groups. In the part of the results, the number of oocytes in particular stages of the experiment was presented.

  1. The oocytes are matured to meiosis II eggs I presume?

Thank you for the suggestions. The oocytes were in the Methapfase II stage, with the first polar body visible. The information was included in the manuscript.

  1. Can the authors comment on the ease of the device used in the vitrification methods? Is the Rapid-I device different from the flat ones? Is it easier to use?

Thank you for the suggestions. All devices are similar: they are all flat, Rapid I devices have an additional hole, but for an experienced laboratory technician, working with them is comparable. Comments have been included.

  1. Please provide info about what the numbers indicate in the tables. If oocyte numbers then please indicate from how many animals/hysterectomy procedures these are collected from. Otherwise, the statistics is hard to evaluate.

Thank you for the suggestion. The number of oocytes is included in Table 2. The number of animals, ovary and oocytes were included in the first part of the results.

Minor points: 

  1. Please check English grammar. This is a minor point but will greatly help to understand the material. 

Thank you for the suggestions. Linguistic corrections have been made.

  1. Please proofread to make sure periods are after every sentence, for e.g. Line 52.

Thank you for the suggestions. The mistake was corrected.

  1. Line 82- I think the authors mean "pooled" and not "pulled".

Thank you for the suggestions. The misstake was corrected.

  1. Line 117 – Should be "plunged" and not "plugged"

Thank you for the suggestions. The mistake was corrected.